# PrgE: an OB-fold protein from plasmid pCF10 with striking differences to prototypical bacterial SSBs

Annika Breidenstein[1,2] , Anaïs Lamy[1,2] , Cyrielle PJ Bader[1] , Wei-Sheng Sun[1,2] , Paulina H Wanrooij[1] , Ronnie P-A Berntsson[1,2]

A major pathway for horizontal gene transfer is the transmission of DNA from donor to recipient cells via plasmid-encoded type IV secretion systems (T4SSs). Many conjugative plasmids encode for a single-stranded DNA-binding protein (SSB) together with their T4SS. Some of these SSBs have been suggested to aid in establishing the plasmid in the recipient cell, but for many, their function remains unclear. Here, we characterize PrgE, a proposed SSB from the *Enterococcus faecalis* plasmid pCF10. We show that PrgE is not essential for conjugation. Structurally, it has the characteristic OB-fold of SSBs, but it has very unusual DNA-binding properties. Our DNA-bound structure shows that PrgE binds ssDNA like beads on a string supported by its N-terminal tail. In vitro studies highlight the plasticity of PrgE oligomerization and confirm the importance of the N-terminus. Unlike other SSBs, PrgE binds both double- and single-stranded DNA equally well. This shows that PrgE has a quaternary assembly and DNA-binding properties that are very different from the prototypical bacterial SSB, but also different from eukaryotic SSBs.

## Introduction

Horizontal gene transfer is an important way for bacteria to spread genetic information between populations, for example, for the propagation of antibiotic resistance or virulence genes (Von Wintersdorff et al, 2016). Conjugation is one type of horizontal gene transfer, which allows for the transfer of plasmids from donor to recipient cells via type IV secretion systems (T4SSs) (Waksman, 2019). These systems are increasingly well understood in Gram-negative bacteria, where recent cryo-EM structures provide an understanding of the mating channel at a molecular level (Macé et al, 2022; Costa et al, 2024). In contrast, our current understanding of Gram-positive T4SSs is much more limited as such detailed information is not available (Grohmann et al, 2018).

One of the best studied Gram-positive T4SSs is from the conjugative plasmid pCF10 (Hirt et al, 2005; Dunny & Berntsson, 2016). This plasmid is a clinical isolate from *Enterococcus faecalis*, a commensal pathogen that often causes hospital-acquired infections and is frequently multiresistant to antibiotics (Palmer et al, 2010; Gilmore et al, 2013; Mikalsen et al, 2015; Weiner-Lastinger et al, 2020). pCF10 is a pheromone-inducible plasmid with a complex regulation (Kohler et al, 2019; Lassinantti et al, 2021). All T4SS proteins on pCF10 are encoded on a single operon, controlled by the $P_Q$ promoter. This operon thus contains the genes that code for (i) some of the regulatory proteins, (ii) the adhesin proteins that facilitate mating pair formation, (iii) the proteins that form the mating channel, and (iv) the DNA transfer and replication (Dtr) proteins, including ATPases and relaxosome proteins (Fig 1) (Dunny, 2013; Grohmann et al, 2018). The relaxosome is made up of an accessory factor PcfF and the relaxase PcfG, which nicks and binds covalently to the origin of transfer and gets transferred together with the single-stranded plasmid DNA into the recipient cell (Guzmán-Herrador & Llosa, 2019; Rehman et al, 2019).

Many conjugative plasmids encode additional proteins that are not directly involved in conjugation, but have various functions that confer competitive advantages to the plasmid (Cooke & Herman, 2023). PrgE is a small soluble protein that is encoded roughly one-third into the $P_Q$ operon, in between genes encoding for the mating channel (Fig 1). PrgE has not been previously characterized, and its role in type IV secretion is therefore unknown, but it has been suggested that PrgE is a single-stranded DNA-binding protein (SSB), based on its sequence homology of 37% to an SSB in a lactococcal phage (Desiere et al, 2001; Hirt et al, 2005).

SSBs are involved in all molecular mechanisms that require manipulation of single-stranded (ss) DNA, such as DNA replication, recombination, and repair, and can be found in all kingdoms of life (Marceau, 2012). Generally, SSBs share a structural motif, the oligosaccharide/oligonucleotide-binding (OB) fold. The motif consists of a five-stranded beta-barrel followed by a single alpha-helix. However, there is a lot of variability in the loops between the beta-strands, the length of OB domains can range from 70 to 150 amino acids, and they often have a low primary sequence identity of

[1]Department of Medical Biochemistry and Biophysics, Umeå University, Umeå, Sweden   [2]Wallenberg Centre for Molecular Medicine and Umeå Centre for Microbial Research, Umeå University, Umeå, Sweden

Correspondence: ronnie.berntsson@umu.se

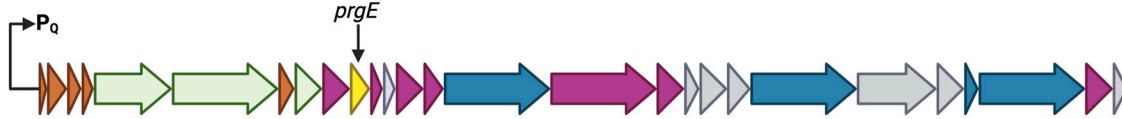

**Figure 1. Schematic overview of the genes included in the P$_Q$ operon of pCF10.**
Each arrow represents one gene, colored by its proposed function in the T4SS. Genes coding for proteins involved in T4SS regulation are shown in orange, surface adhesins in green, mating channel in purple, DNA transfer and replication (Dtr) proteins in blue, and genes of unknown function in gray. The length of the arrows is approximately to scale of the corresponding genes. *prgE* is highlighted in yellow.

5–25% (Theobald et al, 2003; Mishra & Levy, 2015). Although the topology of the OB-fold is well conserved, the quaternary organization of SSBs varies between the different kingdoms of life. The *Escherichia coli* SSB, which is the prototype for bacterial SSBs, forms a homotetramer with two distinct DNA-binding modes, depending on salt and protein concentrations. In the first binding mode, *E. coli* SSB interacts with ssDNA with only two of its subunits, whereas the ssDNA wraps around the full tetramer in the second DNA-binding mode (Lohman & Ferrari, 1994; Raghunathan et al, 2000; Shereda et al, 2008). In eukaryotes, the prototypical SSB is replication protein A (RPA). RPA forms a heterotrimer consisting of RPA70, RPA32, and RPA14, with each subunit containing at least one OB-fold (Liu & Huang, 2016; Nasheuer et al, 2024). When it comes to archaea, some phyla have SSBs that resemble bacterial SSBs, whereas others are more similar to eukaryotic RPA (Taib et al, 2021). There are not only viruses that rely exclusively on host SSBs, but also those that encode their own proteins, with a large diversity of characteristics, some of which act as monomers (Shokri et al, 2009; Oliveira & Ciesielski, 2021). However, there is also variation within the kingdoms, as many bacterial and eukaryotic species have more than one type of OB-fold protein, which can vary significantly from their respective prototypes (Richard et al, 2008; Flynn & Zou, 2010; Yadav et al, 2012; Oliveira & Ciesielski, 2021).

In addition to chromosomal SSBs, many prokaryotes carry conjugative plasmids that encode SSBs (Golub & Low, 1985; Ruvolo et al, 1991). These are believed to contribute to plasmid maintenance, and are thought to be important for protecting ssDNA during conjugation (Ruvolo et al, 1991; Jones et al, 1992; Couturier et al, 2023). Many plasmid SSBs can complement deficiencies in genomic SSBs (Golub & Low, 1985). Recently, it was shown that the F plasmid–encoded T4SS can translocate plasmid SSB into recipient cells where they function to suppress the mating-induced SOS response (Al Mamun et al, 2021). However, it is not known whether SSBs encoded on conjugative plasmids from Gram-positives are functionally analogous.

In this study, we show that PrgE plays no essential role in conjugation, but that it has very unusual DNA-binding properties. Crystal structures of apo and DNA-bound PrgE show that PrgE has the characteristic OB-fold of SSBs, but that it binds ssDNA in a filamentous way, which is further supported by in vitro experiments. We also present data that show that PrgE unexpectedly binds both ssDNA and dsDNA equally well.

## Results

### PrgE is not a homolog of a genome-encoded *E. faecalis* SSB

To compare PrgE with other proteins, we performed sequence-based homology searches. These yielded very little insights,

besides that PrgE is predicted to be an SSB and found only in Enterococci and other related species from the order Lactobacillales. We performed multiple sequence alignment of PrgE with SSBs encoded on the *E. coli* and *E. faecalis* genome (Fig S1A). PrgE only has a very low sequence identity to both sequences (24% to the aligned regions of *E. faecalis* SSB and 19% to *E. coli* SSB). We also created AlphaFold2 models to investigate structural homology. Genomic SSB from *E. faecalis* strongly resembles typical bacterial SSBs, and the model aligns with *E. coli* SSB with an RMSD of 0.59 Å over 83 residues (Fig S1B). In contrast, the PrgE model differs significantly. It superimposes with an RMSD of 5.4 Å over 80 residues to the model of the genome-encoded *E. faecalis* SSB, with differences in the part of the beta-sheet that is involved in DNA binding in typical bacterial SSBs. It also has differences in the N- and C-terminal regions, and contains more alpha-helices than typical OB-folds (Fig S1C). Performing structural homology searches to the AlphaFold2 model of PrgE using Foldseek (Van Kempen et al, 2024) did not yield better information than the sequence-based searches. Top hits in the Protein Data Bank (PDB) database were only distantly related proteins with an OB-fold, with high *E*-values or low TM scores (Table S1). Searching for *E. faecalis* proteins in the AlphaFold database (AFDB50) only resulted in uncharacterized proteins or proteins with low sequence identity to PrgE. This suggests that PrgE differs from previously studied SSBs.

### PrgE has an OB-fold

PrgE was produced in *E. coli* and purified to homogeneity. We solved the crystal structure of apo PrgE to 2.7 Å, using the AlphaFold2 model of PrgE as a template for molecular replacement. The asymmetric unit contained two copies of the protein in the space group P2$_1$2$_1$2$_1$. Both copies were modeled from residues 1–130, with residues 34 and 35 missing in loop 1 of chain A (Fig S2). For both chains, the remaining C-terminal part (residues 131–144) is missing in the density. PISA analysis shows that this dimer has an interface area of 680 Å$^2$, with 9 H-bonds and three salt bridges. The overarching fold of the protein corresponds to an oligosaccharide/oligonucleotide-binding (OB) fold, characterized by five beta-strands that form a beta-barrel with a 1-2-3-5-4-1 topology, which is only partially closed between strands 3 and 5 (Fig 2A). PrgE also has a 42-residue-long region between strands 3 and 4 that forms two alpha-helices of which the first seemingly contributes to the opening in the barrel between strands 3 and 5. The apo structure overall aligns very well with the predicted AlphaFold2 model of PrgE, having an RMSD of 0.48 Å over 113 residues.

We used DALI (Holm, 2020) and Foldseek (Van Kempen et al, 2024) to search the PDB for the closest structural homolog to

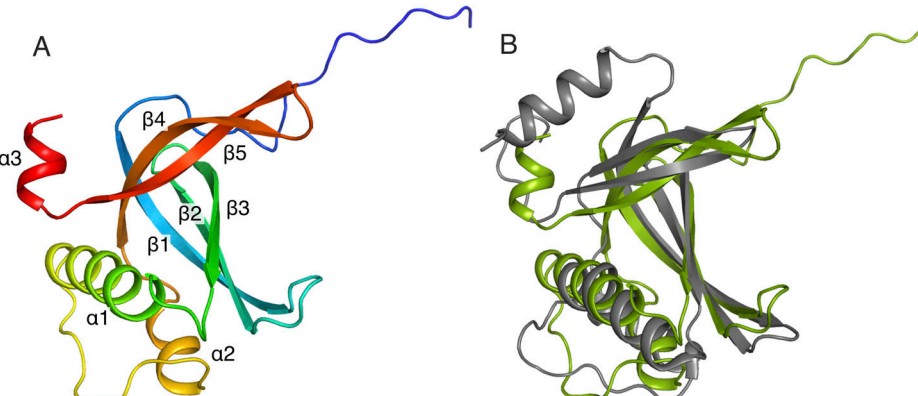

**Figure 2. Apo structure of PrgE.**
**(A)** Crystal structure of PrgE colored in rainbow colors from the N-terminus (blue) to the C-terminus (red). All secondary structure elements are marked in the figure. **(B)** Superimposition PrgE (green) with the C-terminal domain of RadD (gray, PDB: 7R7J). The beta-sheet superimposes relatively well, but there are larger differences in the orientation of the alpha-helices.

PrgE. As with the previous searches with the AlphaFold2 model, the hits had generally very low scores with $E$-values in Foldseek being in the $10^{-2}$ range. The best hit from DALI was the C-terminal domain with an unknown function of the *E. coli* helicase RadD (Osorio Garcia et al, 2022) (PDB: 7R7J) with a Z score of 7.1. However, there are substantial structural differences, which are highlighted by having an RMSD of 4.02 Å over 104 residues between the two structures (Fig 2B).

### PrgE binds ssDNA in a filamentous manner

We also crystallized PrgE together with a single-stranded poly-A 60-mer DNA in a molar ratio of 1:3. The obtained crystallographic data were refined in the space group $P2_12_12_1$ with the asymmetric unit containing three copies of the protein sitting on a string of 15 ssDNA bases. Although there are only 15 bases in the asymmetric unit, the ssDNA shows a continuous density throughout the crystal packing (Fig S3A). Compared with the apo structure of PrgE, a few more residues are visible at the C-terminal end (until residues 136 of 144), continuing as an alpha-helix as predicted by the AlphaFold2 model. The DNA does not get wrapped around PrgE, like it does with *E. coli* SSB (Raghunathan et al, 2000); rather, PrgE interacts with the DNA like beads on a string, with the N-terminal tail of one PrgE binding to the neighboring PrgE, using interactions between polar side chains (Fig 3A). PISA analysis shows that the interaction areas between the PrgE subunits in the DNA-bound structure are between 600 and 800 Å².

PrgE binds to the ssDNA between loops 1 and 4, where the beta-barrel is partially open. Each subunit binds to five DNA bases. The binding also bends the ssDNA between the protein-binding sites, resulting in a kink at every fifth base. The kinks between subunits C'–A and A–B form the same angle. However, the N-terminal tail of chain B bends at a smaller angle and the kink in the DNA chain between subunits B and C is therefore also slightly less pronounced (Fig S3B).

The different PrgE subunits bind to the ssDNA in a similar, but not identical, manner. Many interactions with the phosphate backbone of the ssDNA are the same within all subunits, including with residues Ser33, Gln34, and Asn37 in loop 1 that form H-bonds with the DNA backbone with the fourth and fifth phosphate of each stretch of five bases (Fig 3B–D). Additional phosphate binding can be found with Lys111 and Tyr110 in loop 4 in chains A and C, but not B. Interestingly, this loop interacts with the phosphate of the second base of the DNA-binding cassette that is primarily bound by the neighboring copy of PrgE.

In addition to hydrogen bonding with the phosphate backbone, pi–pi interactions between the aromatic rings of the DNA and two tyrosine residues are of major importance for DNA binding. Tyr110 stacks on the fifth DNA base in the binding cassette in all subunits. In contrast, the orientation of Tyr62 varies. For chains A and B, Tyr62 points inward toward the bases, whereas it is oriented toward the DNA backbone for chain C. Accordingly, the exact orientation of the first DNA base varies between the binding cassettes. In the third binding cassette in the asymmetric unit, base 11 stacks on top of the following four bases and forms two H-bonds with PrgE chain C (Asn120 and Asn66). In the other two cassettes (bound to chains A and B), this base is tilted away and only forms one H-bond with Asn120. Other than these interactions with the DNA bases, hydrogen bonding with DNA bases seems to be less important, consistent with the lack of sequence specificity in DNA binding. In our structure, only Gln108 of chain B interacts with adenine 9, with the other copies of Gln108 being close to the DNA but not in hydrogen bonding distance. In conclusion, PrgE binds to ssDNA with a high degree of plasticity.

### PrgE quaternary structure resembles viral SSBs

The overall quaternary structure of PrgE binding to ssDNA is different than that of bacterial or eukaryotic SSBs, where ssDNA commonly wraps around a homotetramer in bacterial SSBs (Fig 4A) and eukaryotic RPA binds DNA as a heterotrimer (Fig 4B). Instead, it appears more similar to that of viral SSBs, which have monomers as a functional unit in DNA binding (Fig 4C). Each PrgE monomer binds fewer DNA bases (5), which are more neatly stacked on top of each other, compared with other SSBs that have a larger interaction area (Fig 4D–F). The exact DNA-binding mechanisms share some similarities in that stacking interactions with aromatic residues play an important role. However, in PrgE, the responsible residues are tyrosines, whereas they are phenylalanines and tryptophans for *E. coli* SSB and RPA, and the viral SSB uses both tyrosines and phenylalanines.

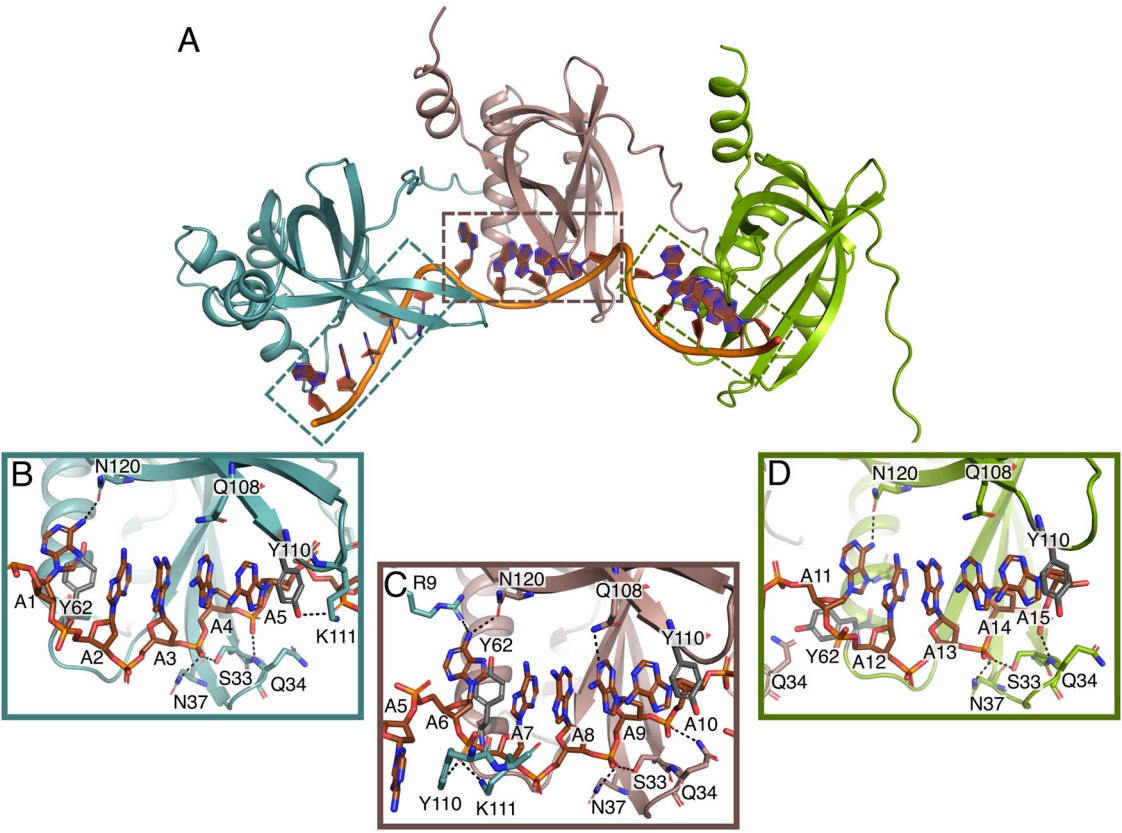

**Figure 3. DNA-bound structure of PrgE.**
**(A)** In the asymmetric unit, there are three PrgE molecules bound to the ssDNA. **(B, C, D)** Enlarged views of the regions indicated in panel (A), highlighting the residues that are important for DNA binding for each of the three monomers. Black dotted lines show potential hydrogen bonds. Orientation of panels (B, C, D) is not the same as in (A), to increase clarity and allow easier comparison.

## The N-terminal tail of PrgE contributes to oligomerization in vitro

Because PrgE oligomerized differently in the crystal structures with or without DNA, we investigated the oligomerization behavior of PrgE in vitro. We noticed that the volume at which PrgE eluted on size-exclusion chromatography (SEC) differed depending on the salt concentration of the buffer (Fig 5A), as well as the protein concentration (Fig 5B). This indicates that PrgE is able to oligomerize. To gain deeper insight into the oligomeric state, we performed size-exclusion chromatography coupled to multi-angle light scattering (SEC-MALS), with 60 $\mu$M PrgE in 300 mM NaCl conditions. The molecular weight of the elution peak was 51.1 ± 2.8 kD, which corresponds well to a trimer (the theoretical molecular weight of the PrgE monomer is 17 kD) (Fig 5C). However, all SEC traces show an asymmetric peak, trailing to the right, indicating the presence of smaller oligomeric species. In addition to this, gentle crosslinking of purified PrgE also captured multiple oligomeric states (Fig 5D). These results show that PrgE can exist in various oligomerization states in vitro and that its oligomerization is both salt concentration– and protein concentration–dependent.

Based on the DNA-bound crystal structure, we hypothesized that the N-terminal tail of PrgE could play an important role in oligomerization. We therefore created a deletion variant where we removed the 12 first residues of PrgE (ΔN-PrgE). This variant eluted

significantly later on SEC than the WT protein; however, we still observed differences in elution volume in different salt concentrations (Fig 6A). To explore these differences in more detail, we performed SEC-MALS in 300 mM NaCl, which resulted in a molecular weight of 16.5 ± 0.6 kD, which is close to the theoretical molecular weight of a ΔN-PrgE monomer (15.5 kD) (Fig 6B). In addition, we performed SEC-MALS in 50 mM NaCl, where ΔN-PrgE was found to form a dimer (molecular weight of 33.1 ± 4.7 kD) (Fig 6C). These results show that the N-terminal tail of PrgE is a major contributor to oligomerization.

## PrgE binds ssDNA and dsDNA with comparable affinities

Given the suggested function of PrgE as an SSB, we performed DNA-binding experiments with both WT and ΔN-PrgE. Binding affinities were compared for random single-stranded (ss) and double-stranded (ds) DNA molecules (Table S2), by determining the dissociation constant ($K_d$) by fluorescence anisotropy (Table 1 and Figs 7 and S4). Surprisingly, PrgE bound ssDNA and dsDNA with similar affinities, with a $K_d$ of 0.3 $\mu$M for 60-mer ssDNA and 0.5 $\mu$M for 60-mer dsDNA in 50 mM NaCl (Fig 7A and B). ΔN-PrgE also bound ssDNA and dsDNA equally well, but it showed a roughly one order of magnitude lower affinity than WT, with 4.5 $\mu$M for 60-mer ssDNA and 5.6 $\mu$M for 60-mer dsDNA (Table 1 and Fig 7A and B). Notably, WT PrgE bound

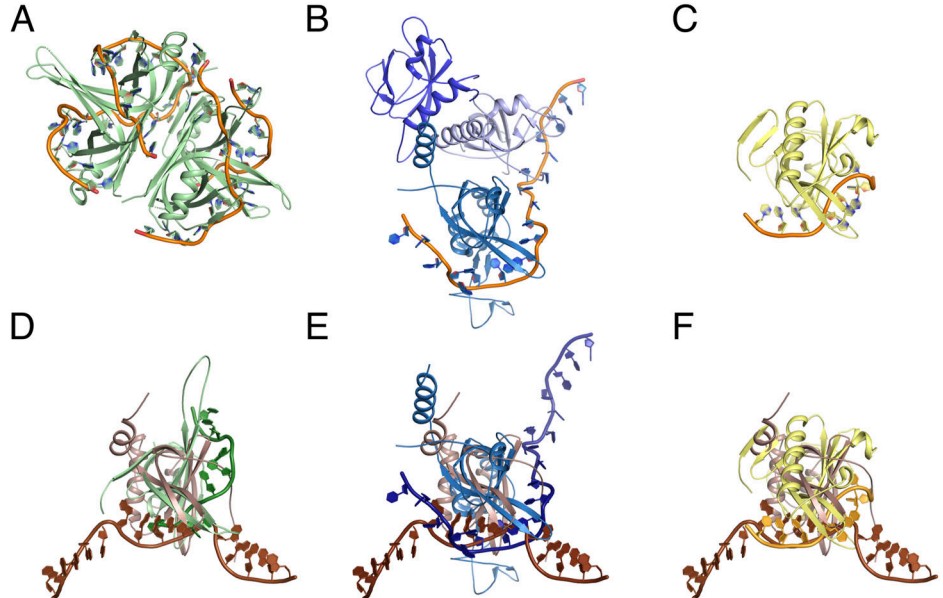

**Figure 4. Comparison between PrgE and other single-stranded DNA-binding proteins (SSBs).** **(A)** *E. coli* homotetrameric SSB bound to ssDNA (PDB: 1EYG). **(B)** Yeast heterotrimeric RPA bound to ssDNA (PDB: 6I52). **(C)** SSB from Enterobacter phage Enc34 (PDB: 5ODL). **(D, E, F)** Superposition of DNA-bound PrgE (brown) with the proteins shown in panels (A, B, C). View in panel (D) is rotated 45° on the x-axis when compared to panel (A) for clarity; the views in panel (E, F) are the same as in (B, C). In panel (E), PrgE is aligned to chain C of RPA as it has the highest structural homology to PrgE.

with higher affinity to the longer DNA substrate, whereas ΔN-PrgE did not show this difference (compare Fig 7A and B with Fig 7C and D). For WT PrgE, we also tested binding in 100 mM NaCl, where the same binding patterns were observed as in lower salt, albeit with somewhat lower affinities (Table 1 and Fig S4A and B). All fluorescence anisotropy data could be fitted using a quadratic equation (Equation (2)) with $R^2 > 0.9$. In addition, we also fitted the data using the Hill equation (Equation (3)), which accommodates cooperativity. For most data, there were no signs of positive cooperativity. However, for PrgE binding to the 60-mer ssDNA, the Hill equation with a Hill coefficient of ca 1.5 fits the data well, suggesting mild positive cooperativity (Fig S4C). This positive cooperativity was not seen with ΔN-PrgE (Fig S4C). All DNA substrates used behaved as expected on agarose gel (Fig S4D). Taken together, these experiments confirm that the DNA-binding properties of PrgE differ considerably from other SSBs, as PrgE binds both ssDNA and dsDNA. They also highlight the importance of the N-terminal tail for DNA binding.

### PrgE is not essential for conjugation

Given that PrgE is a soluble protein in the T4SS operon that binds DNA, we speculated that it might interact with the DNA transfer and replication proteins PcfF (accessory factor [Rehman et al, 2019]) and/or PcfG (relaxase [Chen et al, 2007]), which form the relaxosome at the origin of transfer of plasmid pCF10. We therefore conducted pull-down experiments where untagged PrgE was incubated with either the His-tagged PcfG (Fig 8A) or the GST-tagged PcfF (Fig 8B). However, neither of the proteins co-eluted with PrgE, indicating that they do not strongly interact.

Because PrgE is likely not part of the relaxosome, we wanted to know whether it is essential for conjugation in another way. We therefore created an *E. faecalis* knockout strain (OG1RF:pCF10ΔprgE) to explore the function of PrgE in vivo by comparing the conjugation

efficiency between the mutant and WT. We tested conjugation both during the exponential phase when cells were actively dividing and in the stationary phase when cells are no longer dividing and the availability of other, genome-encoded, SSBs in *E. faecalis* may be different. We observed a decrease in efficiency between exponentially growing cells and cells in the stationary phase, but there was no significant difference between ΔprgE and WT in either condition (Fig 9). We further considered whether multiple conjugative events would be needed to observe an effect. We therefore passaged the plasmids several times between donor and recipient cells, using transconjugant cells as new donor cells. However, also here we did not observe any difference within four passages between ΔprgE and WT (Fig 9). We conclude that PrgE does not play an essential role in conjugation under the tested conditions.

## Discussion

Many conjugative plasmids, with different incompatibility groups, encode for (at least) one SSB protein, which can often complement the genome-encoded SSB (Golub & Low, 1985). In conjugation, SSBs have been proposed to be important for protecting plasmid ssDNA both in donor and in recipient cells and to evade the SOS response (Howland et al, 1989; Jones et al, 1992; Al Mamun et al, 2021; Couturier et al, 2023). However, all of the available research has been done on SSBs from Gram-negative T4SSs. Here, we characterized the proposed SSB PrgE from the Gram-positive conjugative plasmid pCF10.

By crystallizing PrgE, we showed that it indeed has the typical OB-fold of SSBs, but that its structure has important differences when compared to other SSB proteins. PrgE has three alpha-helices that are positioned differently from other SSBs, and also differs in its beta-sheet where the DNA-binding regions are. The differences became even more apparent when we analyzed the DNA-bound

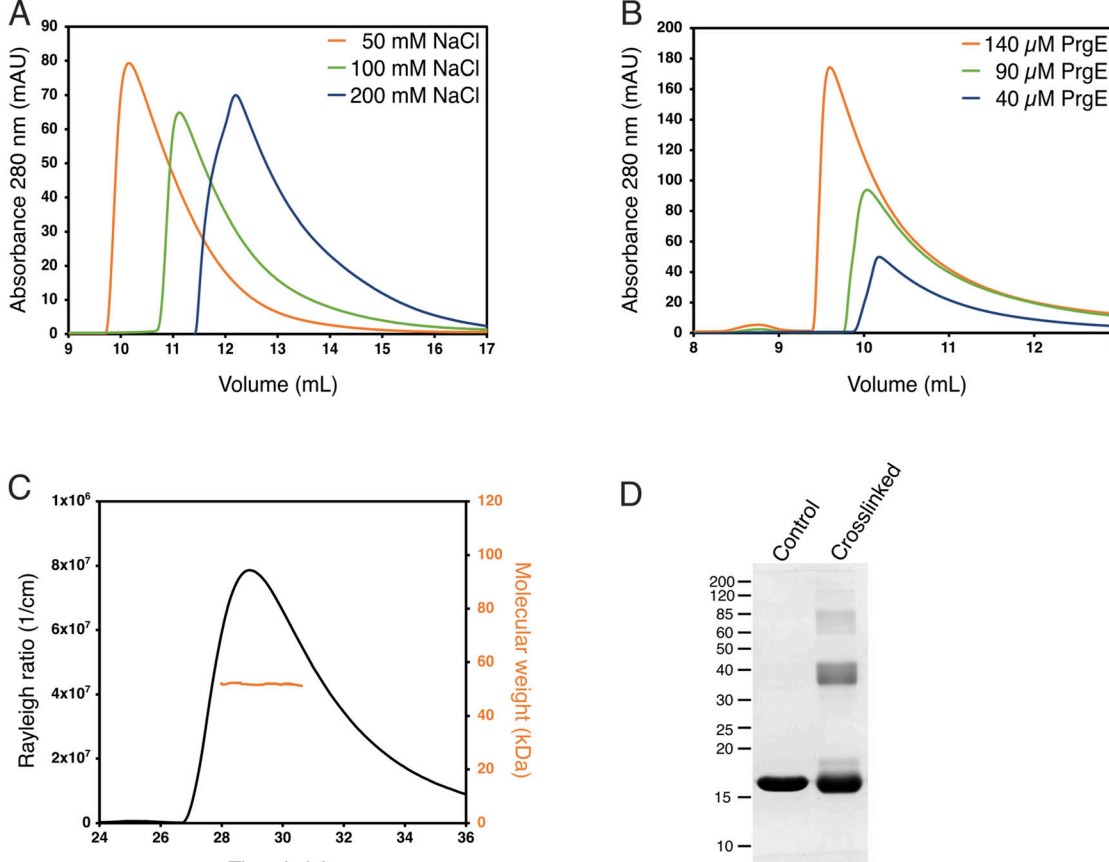

**Figure 5. Oligomerization of PrgE.**
**(A)** Size-exclusion chromatogram of PrgE (on a Superose 6 column) shows that the elution volume, which is coupled to protein radius, depends on the salt concentration. **(B)** Size-exclusion chromatogram of PrgE (on a Superdex 200 column), in the same salt concentration but with different protein concentrations, shows that the elution volume decreases with increasing protein concentrations. **(C)** SEC-MALS analysis of 60 $\mu$M PrgE in 300 mM NaCl. The black line, plotted on the left axis, indicates the Rayleigh ratio, which is directly proportional to the intensity of the scattered light in excess of the buffer. The orange line, plotted on the right axis, indicates the molecular weight of the protein measured throughout the peak. The average molecular weight was 51.1 ± 2.8 kD. **(D)** SDS–PAGE of PrgE, with or without crosslinking with disuccinimidyl suberate.
Source data are available for this figure.

structure. Each monomer binds DNA in a way that is to be expected, relying on interactions with the DNA backbone and stacking interactions with the bases to achieve DNA binding in a sequence-independent manner. However, PrgE does not bind DNA as the typical bacterial SSB, which commonly forms homotetramers around which they wrap the ssDNA. It is also very different from how eukaryotic SSBs, like RPA, bind the ssDNA as heterotrimers. Instead, PrgE binds the ssDNA in a filamentous manner, like beads on a string (Fig 3). Between each binding site, the DNA gets bent (Fig S3B). Whether the exact angles are due to crystal packing or are also the ones found in solution is not known. The oligomerization in the DNA-bound structure is supported by the N-terminal tail of PrgE, which interacts with the neighboring monomer on the DNA-bound structure (Fig 3), a feature that is not found on the prototypical bacterial SSBs. Further supporting the filamentous oligomerization are the different oligomerization states that were observed for PrgE in solution (Fig 5). The N-terminally truncated variant of PrgE (ΔN-PrgE), which was predominantly monomeric and showed capacity to dimerize only in low salt conditions, confirms the role of the

N-terminus in oligomerization that was suggested by the DNA-bound crystal structure (Fig 6).

Most of our data from the fluorescence anisotropy experiments fit best to a standard quadratic binding curve that does not account for cooperativity (Figs 7 and S4). However, for the single-stranded 60-mer substrate, the Hill equation with a positive Hill coefficient fits the data well and indicates cooperativity in the binding (Fig S4C). This cooperative binding was lost for ΔN-PrgE, suggesting that the N-terminal tail does promote cooperative binding on longer DNA substrates. Surprisingly, we found that PrgE bound dsDNA equally well as ssDNA (Figs 7 and S4 and Table 1). Most characterized SSBs have a high affinity and specificity for ssDNA (Oliveira & Ciesielski, 2021). As an example, RPA binds mixed ssDNA with affinities of 10–40 nM albeit displaying a preference for pyrimidines, and with $K_D$ values to ssDNA up to three orders of magnitude lower than to dsDNA (Brill & Stillman, 1989; Wold et al, 1989; Kim et al, 1992). To our knowledge, only one studied SSB-like protein shares PrgE's feature of binding equally well to both ssDNA and dsDNA, namely, one from the archaea *Nanoarchaeum equitans* (Olszewski

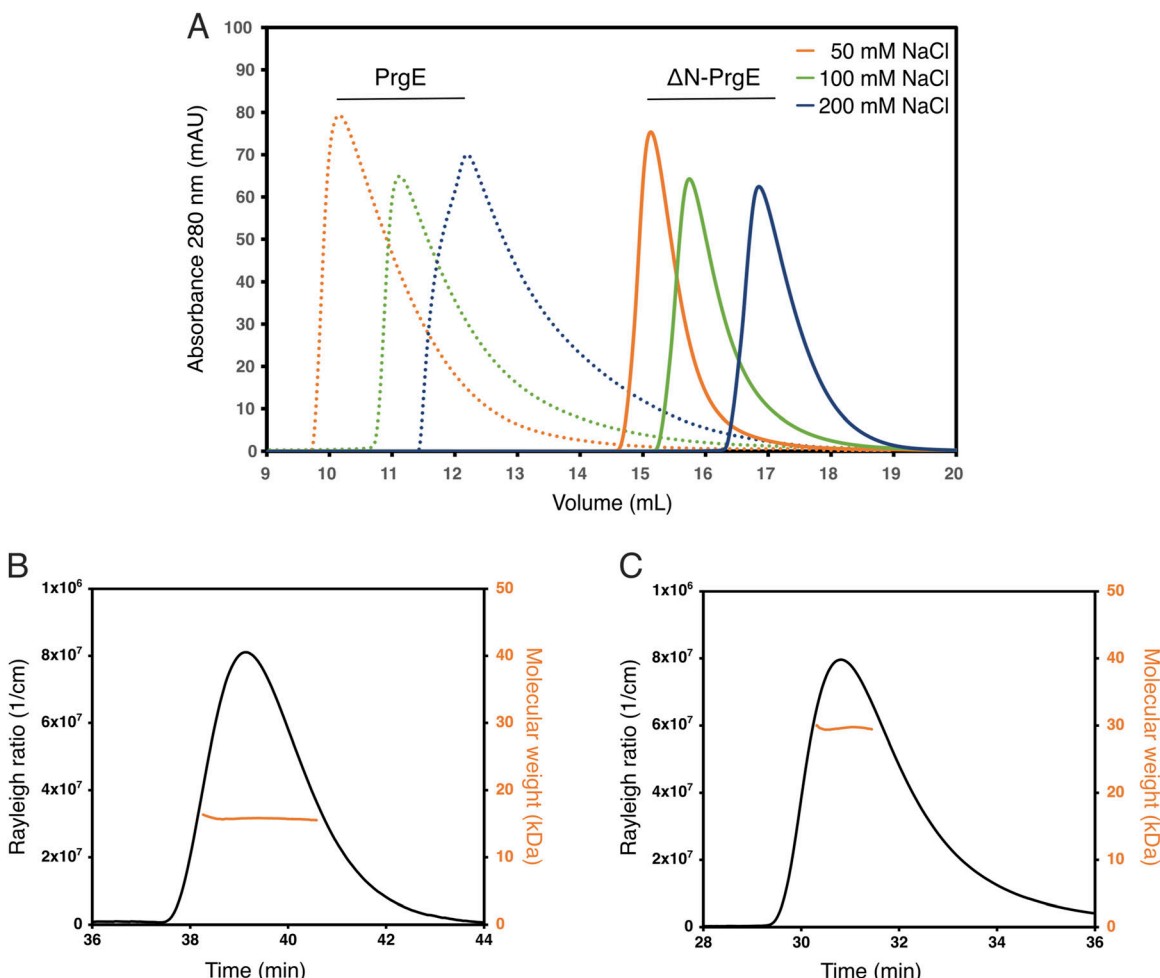

**Figure 6. Oligomerization of ΔN-PrgE.**
**(A)** ΔN-PrgE (solid lines) elutes significantly later than WT (dotted lines, the same as in Fig 5B) on size-exclusion chromatography, but still, its elution volume is dependent on the salt concentrations. **(B)** SEC-MALS analysis of ΔN-PrgE in 300 mM NaCl with the Rayleigh ratio indicated in black on the left axis and the molecular weight in orange on the right axis. The calculated weight was 16.4 ± 0.6 kD, which is close to that of a monomer. **(C)** SEC-MALS analysis of ΔN-PrgE in 50 mM NaCl gave a calculated molecular weight of 33.1 ± 4.7 kD, which is close to that of a dimer.

et al, 2015). When PrgE binds dsDNA, the DNA must be in a different conformation than in our ssDNA-bound structure. This makes it difficult to speculate exactly how PrgE would structurally bind dsDNA, besides that the residues interacting with the ssDNA phosphate backbone likely also are important for dsDNA binding. Given these data, it is clear that PrgE is not a typical SSB, and we therefore refer to it simply as an OB-fold protein.

Given these unexpected characteristics of PrgE, it is tempting to speculate about its evolutionary origin. Despite being present in the middle of a T4SS operon on a bacterial conjugative plasmid, PrgE does not behave at all like a bacterial SSB. No close structural homologs could be identified via DALI (Holm, 2020) and Foldseek (Van Kempen et al, 2024). PrgE's oligomerization behavior in DNA binding, where PrgE monomers can be added like beads on a string in a non-cooperative manner, is reminiscent of some viruses whose SSBs have a monomer as a functional subunit that can be added on ssDNA (Dekker et al, 1997; Shokri et al, 2009). We did find similarities regarding DNA-binding affinities with an archaeal SSB, which is

described as resembling viral SSB-like proteins (Olszewski et al, 2015; Oliveira, 2021). Indeed, the C-terminally truncated Enc34 phage SSB has been shown to bind dsDNA (Cernooka et al, 2017). Furthermore, the Enc34 SSB was also suggested to be able to bind DNA in a filamentous manner, similar to what we here observe for PrgE (Cernooka et al, 2017). In addition, PrgE was originally annotated as an SSB protein based on its 37% sequence similarity to a lactococcal phage SSB (Desiere et al, 2001). We therefore find it likely that PrgE at some point has been introduced to pCF10 via horizontal gene transfer mediated by a phage.

What then is the function of PrgE for the T4SS and in conjugation? PrgE is expressed as part of the $P_Q$ operon of pCF10, surrounded by proteins that are essential for its T4SS (Fig 1). This means that PrgE will be produced only when transcription of the $P_Q$ operon has been induced, and its production will be quickly shut down again, just like the rest of the proteins encoded by the $P_Q$ operon (Lassinantti et al, 2021). Our first hypothesis was that PrgE might interact with other important DNA-binding components of type IV secretion, the

**Table 1. $K_d$ values and standard deviations (n = 3) for PrgE and ΔN-PrgE binding to ssDNA or dsDNA oligonucleotides in 50 or 100 mM NaCl as determined by fluorescence anisotropy using Equation (2) (quadratic fit).**

| $K_d$ ± SD ($\mu$M) | | ssDNA | | dsDNA | |
|---|---|---|---|---|---|
| | | 30-mer | 60-mer | 30-mer | 60-mer |
| 50 mM NaCl | WT PrgE | 1.02 ± 0.02 | 0.33 ± 0.02 | 1.96 ± 0.26 | 0.50 ± 0.14 |
| | ΔN-PrgE | 3.85 ± 0.49 | 4.49 ± 0.48 | 4.39 ± 0.24 | 5.56 ± 0.19 |
| 100 mM NaCl | WT PrgE | 1.84 ± 0.15 | 0.37 ± 0.01 | 3.36 ± 0.28 | 0.95 ± 0.03 |

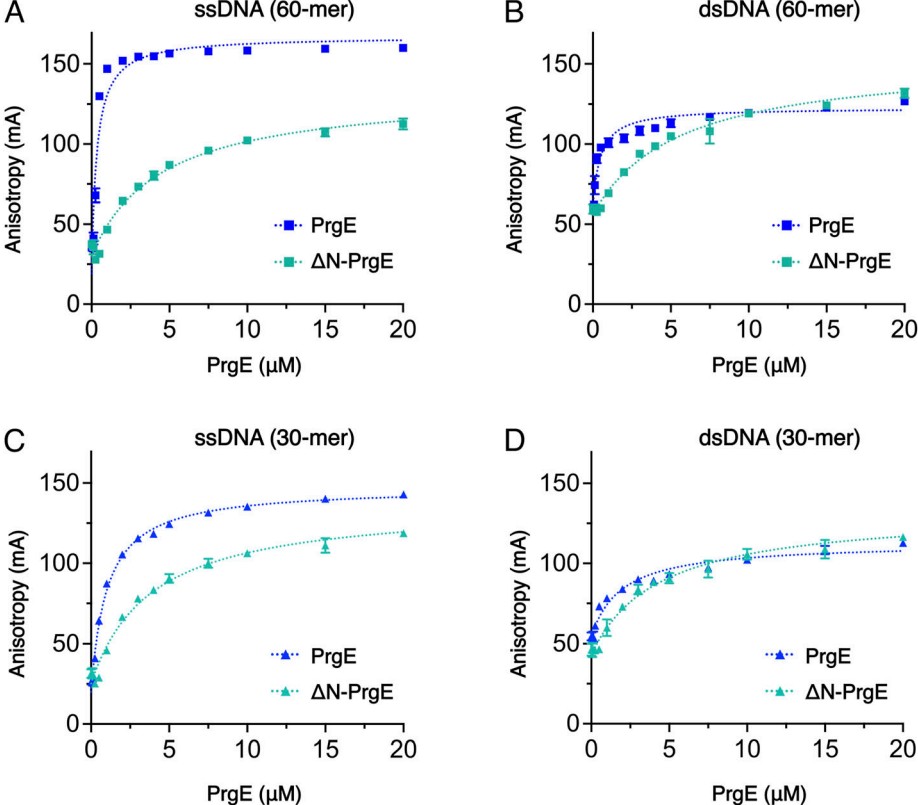

**Figure 7. PrgE DNA binding.**
Fluorescence anisotropy measurements of PrgE (blue) and ΔN-PrgE (green) with random DNA in reaction buffer containing 50 mM NaCl. **(A)** Binding to a single-stranded DNA (ssDNA) 60-mer substrate. **(B)** Binding to a double-stranded DNA (dsDNA) 60-mer substrate. **(C)** Binding to a 30-mer ssDNA substrate. **(D)** Binding to a 30-mer dsDNA substrate. Error bars (only visible when larger than the symbol) represent the SD (n = 3). Curves were fitted with Equation (2) (quadratic fit).

relaxosome proteins PcfG and PcfF, as SSBs can be important players in recruiting proteins to DNA (Bianco, 2017; Antony & Lohman, 2019). However, PrgE does not seem to interact strongly with either of them. Secondly, we speculated that PrgE was important for conjugation in other ways, potentially by protecting the conjugative ssDNA in either the donor or recipient strain, or maybe by aiding the establishment of the plasmid in the recipient cells (Couturier et al, 2023). To test this, we created a knockout of PrgE (pCF10:ΔprgE). However, no significant differences in conjugation efficiency could be observed, neither in the exponential phase nor in the stationary phase. It also did not affect the efficiency during multiple serial conjugation events. This is in line with what was observed in previous studies on an F-plasmid, where knocking out a plasmid-encoded *ssb* also did not reduce mating rates (Al Mamun et al, 2021). However, these experiments were performed under laboratory conditions, and it is possible that PrgE does contribute to conjugation efficiency under other, less ideal, circumstances.

Conjugative plasmids retain many proteins that are not strictly required for conjugation itself, but provide various other advantages, for example, competitiveness against other conjugative elements or replacement of host functions that allows plasmids to use a wider host range (Cooke & Herman, 2023). The F-plasmid encodes an SSB that gets transferred into the recipient cell where it suppresses the SOS response (Al Mamun et al, 2021). It could be one potential avenue to explore whether also PrgE can be transferred through the T4SS and serve a similar function in the *E. faecalis* recipient cell. However, we deem it unlikely that PrgE has a homologous function, given that the F-plasmid SSB is a typical bacterial SSB that can compensate for genomic SSB deficiencies (Chase et al, 1983; Kolodkin et al, 1983), whereas PrgE is very different from *E. faecalis* SSB and has very unusual DNA-binding characteristics. In addition, it has yet to be demonstrated whether the pCF10 T4SS can transfer proteins other than DNA-coupled relaxases. The ability of PrgE to bind both ssDNA and

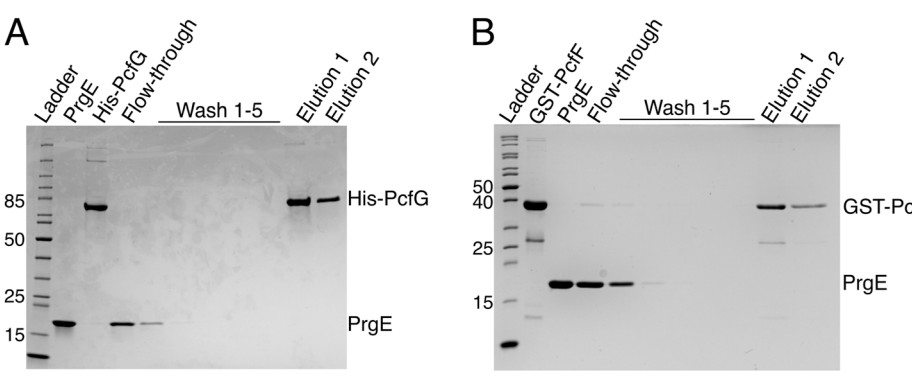

**Figure 8.  PrgE does not interact with the main components of the pCF10 relaxosome.**
**(A)** Pull-down experiment with the relaxase PcfG, showing the input protein, washes, and elution, in which His-PcfG (bait) was unable to pull down PrgE (prey). **(B)** Pull-down experiment in which the relaxosome accessory factor GST-PcfF (bait) was unable to pull down PrgE (prey).
Source data are available for this figure.

dsDNA increases the range of potential functions to any cellular process involving DNA. Understanding the exact function of PrgE remains an exciting prospect for future research.

Conjugative plasmids have been studied for many decades now, ever since the R1 conjugative plasmid was first isolated from a clinical isolate in 1963 (Datta & Kontomichalou, 1965). Genes encoding for OB-fold proteins are part of these plasmids, but our understanding of their specific function within conjugation remains very limited and is almost exclusively based on T4SSs from Gram-negative bacteria. Here, we have shown that PrgE from the Gram-positive conjugative plasmid pCF10 behaves differently to the more well-studied SSBs. It binds ssDNA by attaching PrgE monomers to the DNA like beads on a string, instead of around a globular oligomer like *E. coli* SSB, and it binds dsDNA equally well as ssDNA. Its oligomerization behavior and DNA-binding mechanism are instead providing insight into a class of OB-fold proteins that has been very poorly characterized.

## Materials and Methods

### Cloning, plasmids, and strains

Strains, oligos, and plasmids used in this study are listed in Table S2. *E. coli* strains were cultured in Lysogeny Broth (LB) or Terrific Broth (TB) supplemented, when necessary, with antibiotics at the following concentrations: 100 μg/ml kanamycin, 20 μg/ml genta-mycin, and 25 μg/ml chloramphenicol. *E. faecalis* strains were cultured in Brain–Heart Infusion (BHI) Broth or Tryptic Soy Broth without Dextrose (TSB-D) supplemented, when necessary, with antibiotics at the following concentrations: 10 μg/ml chloram-phenicol, 10 μg/ml tetracycline, 25 μg/ml fusidic acid, and 20 μg/ml erythromycin.

The sequence encoding *prgE* was PCR-amplified from the pCF10 plasmid using primers PrgE_FX_F or ΔN-PrgE_FX_F and PrgE_FX_R and cloned into the intermediate vector pINIT_kan after digestion by *SapI*, using the FX cloning system (Geertsma & Dutzler, 2011). It was subcloned into the expression vector p7XC3H, which provides a C-terminal 10xHis-tag and a 3C protease cleavage site, before transformation of *E. coli* ArcticExpress (DE3) cells. The sequence encoding *pcfG* was PCR-amplified using the primers PcfG_F and PcfG_R and cloned into a pET24d vector after digestion with *Eco31I*,

which provides an N-terminal 10xHis-tag and a SUMO-tag, before transformation into *E. coli* BL21 (DE3) cells.

The *E. faecalis* PrgE-deleted strain, OG1RF:pCF10Δ*prgE*, was obtained by allelic exchange and counter-selection using a pCJK218 plasmid (Vesić & Kristich, 2013), leaving the nucleotides encoding the first and last five amino acids of the protein. About 800 bp of the upstream and downstream regions of PrgE was PCR-amplified using the primer pairs PrgE-UF-F/PrgE-UF-R and PrgE-DF-F/PrgE-DF-R, respectively. The products were digested by *BamHI/SalI* for the upstream region and *SalI/NcoI* for the downstream region, before cloning into the pCJK218 digested by *BamHI/NcoI*. The resulting plasmid was used to transform *E. faecalis* OG1RF:pCF10 by electroporation (Bae et al, 2002). The PrgE-deleted transformants were obtained by switching temperature to induce allelic exchange as described by Vesić and Kristich (2013), and the gene deletion was subsequently confirmed by sequencing.

### Protein production

Proteins were expressed using the LEX system (Large-scale EX-pression system, Epiphyte 3). PrgE and ΔN-PrgE were transformed in *E. coli* ArcticExpress (DE3) cells and cultivated in TB medium supplemented with 0.4% glycerol. The cultures were grown at 30°C until an $OD_{600}$ of 0.8, then cooled down to 12°C before 0.4 mM IPTG was added to induce protein expression. After 24 h, cells were centrifuged at 4,000$g$ during 20 min. PcfF was produced the same way, with the exception that BL21 (DE3) cells were used, and cultures were grown at 37°C before lowering the temperature to 18°C before induction, and harvested after 20 h. PcfG was produced in Origami (DE3) cells using autoinduction TB media. Cultures were grown at 37°C until OD 0.6 was reached, followed by 24 h at 25°C without the addition of IPTG.

### Protein purification

Cell pellets were resuspended in different lysis buffers. For PrgE and ΔN-PrgE, this lysis buffer consisted of 50 mM Tris–HCl, pH 8, 10% glycerol, 500 mM NaCl, 10 mM imidazole, 0.2 mM AEBSF, 1 mM DTT, 1 mM MgSO₄, and 0.02 mg/ml DNase I. For PcfF, the lysis buffer was 50 mM Hepes, pH 7.5, 500 mM NaCl, 0.2 mM AEBSF, and 0.02 mg/ml DNase I. For PcfG, the lysis buffer was 50 mM Tris–HCL, pH 8, 10% glycerol, 500 mM NaCl, 10 mM imidazole, 0.2 mM AEBSF, and 0.02 mg/

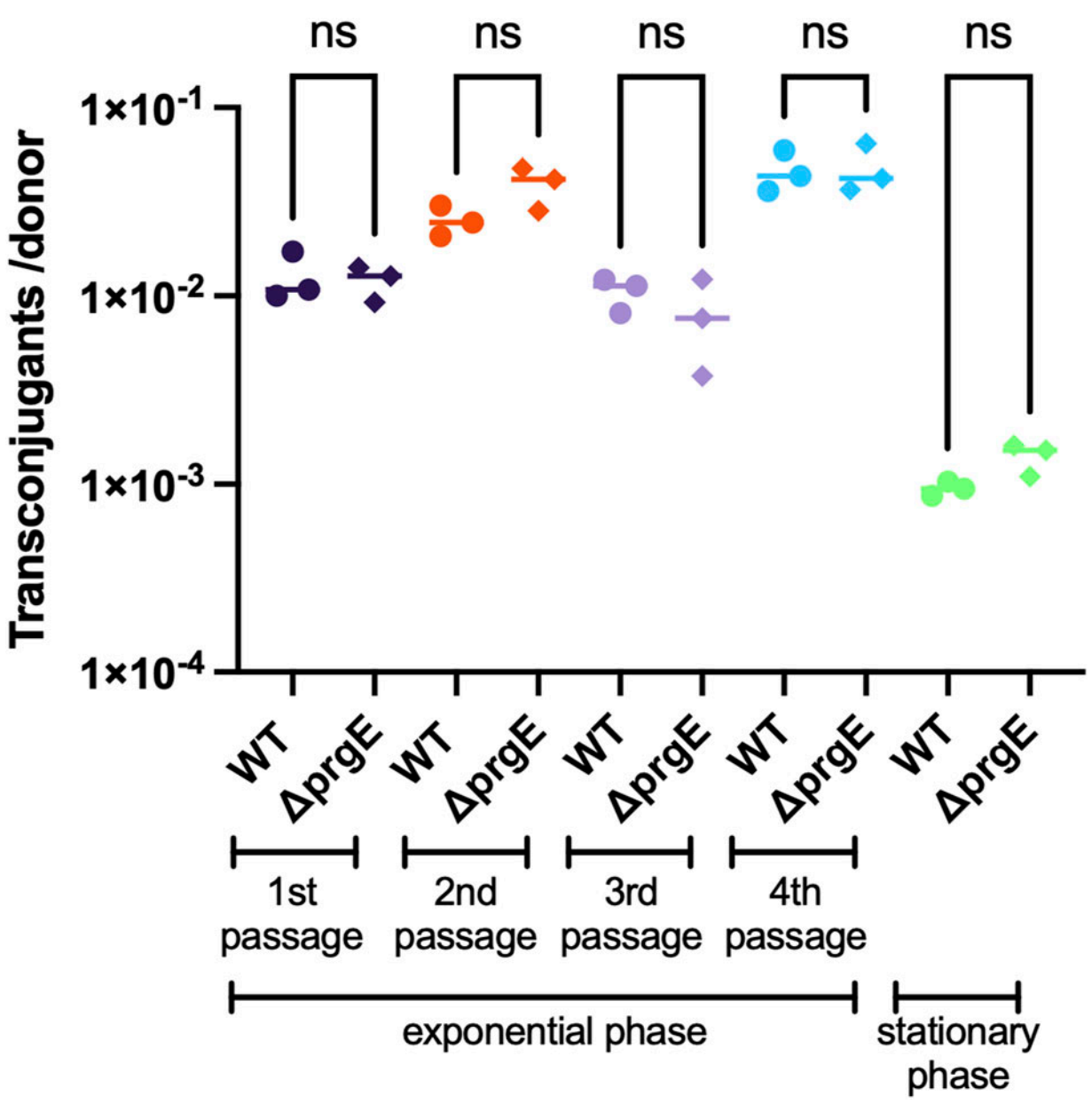


**Figure 9. PrgE is not essential for conjugation.**
Conjugation rates of *E. faecalis* donor cells carrying WT pCF10 or pCF10:Δ*prgE* either in the exponential phase or in the stationary phase. In the exponential phase, serial passaging was performed, where transconjugants from one passage were used as donor cells in the following passage. ns stands for not significant.

ml DNase I. Resuspended cells were lysed in Cell Disruptor (Constant Systems) at 25 kPsi and centrifuged at 30,000*g* for 30 min at 4°C.

The PrgE-His or ΔN-PrgE-His supernatant was incubated for 1 h at 4°C with gentle rocking with Ni-NTA resin (Protino). After incubation, the mix was transferred into a gravity flow column and washed with three subsequent 20 CV washes with wash buffer (20 mM Tris–HCl, pH 7.5, 5% glycerol, 500 mM NaCl, 50 mM imidazole), LiCl wash buffer (wash buffer supplemented with 2 M LiCl), and wash buffer. The resin was then incubated overnight at 4°C with gentle rocking in elution buffer (20 mM Tris–HCl, pH 7.5, 500 mM NaCl, 5% glycerol, 50 mM imidazole, 1 mg of PreScission protease, 1 mM DTT), during which the His-tag was cleaved. The flow-through, as well as an additional wash with 5 CV elution buffer, was collected and concentrated using Amicon Ultra Centrifugal filters with a molecular weight cutoff of 10 or 3 kD for ΔN-PrgE. The protein was then subjected to SEC in 20 mM Hepes, pH 7.5, and 300 mM NaCl on a HiLoad 16/600 Superdex 200 pg column using ÄKTA Purifier (Cytiva). For buffer exchange before various experiments, PrgE or ΔN-PrgE was subjected to a second SEC using a Superose 6 Increase 10/300 GL column on ÄKTA pure (Cytiva).

The GST-PcfF supernatant was incubated for 1 h with glutathione resin (GE Healthcare) at 4°C and subsequently washed with 50 CV wash buffer (20 mM Hepes, pH 7.5, 200 mM NaCl) before elution with 20 mM Hepes, pH 7.5, 200 mM NaCl, 30 mM glutathione. The protein was concentrated with Amicon Ultra Centrifugal filters with a molecular weight cutoff of 10 kD before SEC in 20 mM Hepes, pH 7.5, 200 mM NaCl on a Superdex 200 Increase 10/300 GL column using ÄKTA pure (Cytiva).

The His-PcfG supernatant was incubated for 1 h at 4°C with pre-equilibrated HisPur Cobalt Resin (Thermo Fisher Scientific) to three subsequent 20 CV washes with wash buffer (20 mM Hepes, pH 7.5 5% glycerol, 300 mM NaCl, 30 mM imidazole), LiCl wash buffer (wash buffer supplemented with 2 M LiCl), and wash buffer. PcfG was then eluted in 15 CV elution buffer (20 mM Hepes, pH 7.5, 5% glycerol, 300 mM NaCl, 150 mM imidazole). The protein was loaded onto a HiTraP Heparin HP (5 ml) column (GE Healthcare) equilibrated with Buffer A (20 mM Hepes, pH 7.5, 150 mM NaCl). PcfG was eluted in a salt gradient to 100% Buffer B (20 mM Hepes, pH 7.5, 1,000 mM NaCl).

## Crystallization and structure determination

SEC-purified PrgE, with a concentration of 11 mg/ml, was used for crystallization trials. Crystals appeared after 2–5 d, at 20°C, using the vapor diffusion method in a condition with 0.2 M $LiSO_4$, 0.1 M K Phos Cit, pH 4.2, 20% wt/vol PEG 1000 in a 2:1 ratio. For the DNA-bound structure, 117 μM of single-stranded poly-A 60-mer was added to 6 mg/ml PrgE and mixed in a 1:2 ratio with a reservoir solution containing 15% vol/vol PEG 400, 50 mM MES, pH 6.5, 80 mM Mg acetate, 15 mM $MgCl_2$. Crystals were flash-frozen in liquid nitrogen without an additional cryoprotectant. X-ray diffraction data were collected at the ID30A-3 (apo) or ID23-1 (DNA-bound) beamlines at the ESRF, France, and processed using XDS (Kabsch, 2010). The space group of both crystals was $P2_12_12_1$, and the phase problem was solved in Phenix Phaser (McCoy et al, 2007) using molecular replacement with an AlphaFold2 (Jumper et al, 2021) model of PrgE where the flexible extremities of the protein had been removed, generated using ColabFold version 1.5.2 using default settings (Mirdita et al, 2022). The asymmetric unit of the crystal contained two copies of PrgE for the apo structure. The asymmetric unit of the DNA-bound protein contained three copies of the protein and a 15-nucleotide stretch of the single-stranded DNA. The chosen asymmetric unit thus contains only a quarter of the full ssDNA that the protein was crystallized with. We chose to do so because the ssDNA has continuous density throughout the crystal packing, and this greatly simplified the refinement process. The structures were built in Coot (Emsley & Cowtan, 2004) and refined at 2.7 Å using Refmac5 (Vagin et al, 2004), and we obtained $R_{work}/R_{free}$ values of 23.45 and 27.77 for the apo structure and 23.05 and 25.23 for the DNA-bound structure. Further refinement statistics can be found in Table S3.

## SEC-MALS

For analysis of the oligomeric state of PrgE, 150–300 μl of 1 mg/ml PrgE or ΔN-PrgE (with a theoretical mass of 17 or 15.5 kD, respectively) was loaded on a Superdex 200 Increase 10/300 GL column, equilibrated in buffer (20 mM Hepes, pH 7.5, and 300 mM NaCl) via ÄKTA pure (Cytiva) that was coupled to a light scattering (Wyatt Treas II) and refractive index (Wyatt Optilab T-Rex) detector to determine the molecular weight of the elution peak via SEC-MALS. Data were analyzed using Astra software (version 7.2.2; Wyatt Technology).

## Crosslinking

PrgE crosslinking experiments were performed by incubating 30 μg of protein with 2 mg of disuccinimidyl suberate in 20 mM Hepes, pH

7.5, and 300 mM NaCl for 30 min at 20°C. The reaction was quenched by adding 100 mM Tris–HCl, pH 8.0, at least 10 min before analysis using SDS–PAGE with Coomassie Brilliant Blue staining.

### Preparation of DNA substrates

Oligonucleotides were purchased from Eurofins and are listed in Table S2. For double-stranded substrates, one nmol of each oligonucleotide was annealed to an equimolar amount of its complementary strand by denaturing at 95°C for 5 min in TE buffer (50 mM Tris–HCl, pH 8.0, 1 mM EDTA) containing 100 mM NaCl, and allowing the reaction mixture to cool to RT. The DNA was separated on a 15% acrylamide gel in 0.5 × TBE (15 mM Tris, 44.5 mM boric acid, 1 mM EDTA), stained with 3 × GelRed (Biotium) for 30 min, and visualized using ChemiDoc (Bio-Rad). The bands corresponding to double-stranded molecules were excised with a clean razor blade, eluted from crushed gel slices into TE buffer (10 mM Tris–HCl, pH 8.0, 1 mM EDTA), and purified by phenol–chloroform extraction and isopropanol precipitation.

### Fluorescence anisotropy assay

Single-stranded and double-stranded oligonucleotides of 30 or 60 nt with a 5′ FITC label (Table S2) were diluted to 20 nM in binding buffer (20 mM Hepes, pH 7.5, 50 or 100 mM NaCl, as indicated). Before use, the single-stranded oligonucleotides only were boiled for 5 min at 95°C and chilled on ice. Fluorescence anisotropy reactions containing 10 nM oligonucleotide and 0–20 $\mu$M PrgE or $\Delta$N-PrgE in binding buffer were pipetted in duplicates onto black, shallow 384-well microplates (OptiPlate-F, PerkinElmer) and incubated in the dark for 30 min at RT. Fluorescence intensities were collected from above on a CLARIOstar *Plus* plate reader (BMG Labtech) with the excitation and emission wavelengths 480 and 520 nm, respectively. Fluorescence anisotropy in millianisotropy units (mA) was calculated using MARS Data Analysis Software (BMG Labtech) according to Equation (1):

$$\text{Fluorescence anisotropy} = \frac{F\parallel - F\perp}{F\parallel + 2 \times F\perp} \times 1000 \tag{1}$$

where F $\parallel$ and F $\perp$ are the parallel and perpendicular emission intensity measurements corrected for background (buffer). PrgE alone exhibited no fluorescence. The dissociation constant ($K_d$) was determined by fitting data to a quadratic equation by non-linear regression analysis in GraphPad Prism software (GraphPad Software, Inc.) using Equation (2):

$$Y = B_0 + (B_{max} - B_0) \times \frac{\sqrt{(D + X + K_d)^2 - (4 \times D \times X)}}{2 \times D} \tag{2}$$

where Y is the anisotropy value at protein concentration X, X is the concentration of PrgE in $\mu$M, $B_0$ and $B_{max}$ are specific anisotropy values associated with free DNA and total DNA-PrgE complex, respectively, and D is the concentration of DNA in $\mu$M.

For 60-nt ssDNA, the data were in addition fitted to the Hill equation by non-linear regression analysis in GraphPad Prism software (GraphPad Software, Inc.) using Equation (3):

$$Y = \frac{B_{max} \times X^h}{K_d{}^h \times X^h} \tag{3}$$

where Y is the anisotropy value at protein concentration X, X is the concentration of PrgE in $\mu$M, $B_{max}$ is the specific anisotropy value associated with total DNA-PrgE complex, and h is the Hill coefficient.

### Pull-down experiments with relaxosome components

PrgE pull-down experiments were performed in 20 mM Hepes, pH 7.5, and 200 mM NaCl by mixing either 2 nmol GST-PcfF or PcfG-His (baits) with 4 nmol PrgE without tag (prey) and 100 $\mu$l of the resin (glutathione resin [GE Healthcare] when using PcfF and Ni-NTA [Protino] for PcfG). The proteins were incubated for 15 min at 4°C before collecting the flow-through and washing with 5 × 5 CV wash buffer and eluting with 2 × 5 CV elution buffer. For GST-PcfF pull-downs, 20 mM Hepes, pH 7.5, and 200 mM NaCl were used as wash buffer and 20 mM Hepes, pH 7.5, 200 mM NaCl, and 30 mM glutathione as elution buffer. For His-PcfG pull-downs, wash buffer contained 20 mM Hepes, pH 7.5, 200 mM NaCl, 30 mM imidazole, and elution buffer, 20 mM Hepes, pH 7.5, 200 mM NaCl, 500 mM imidazole. The samples were analyzed on SDS–PAGE and stained with Coomassie Brilliant Blue.

### Conjugation assays

Donor (OG1RF:pCF10 or OG1RF:pCF10$\Delta$*prgE*) and recipient (OG1ES) strains were inoculated with the indicated antibiotics and incubated overnight at 37°C with agitation. The next day, the overnight cultures were refreshed in BHI media without antibiotics in a 1:10 ratio. For conjugation assays in the exponential phase, cells were directly induced to express the T4SS with 5 ng/ml cCF10 for 1 h at 37°C without agitation. For conjugation assays in the stationary phase, cultures were first incubated for 3 h at 37°C with agitation before induction. Donor and recipient cells were then gently mixed in a 1:10 ratio and incubated for 30 min at 37°C without agitation. To disrupt the ongoing conjugation, cells were vortexed and placed on ice for 10 min. A serial dilution was performed with cold media, and 10 $\mu$l of the appropriate dilutions was spotted in triplicates on the top of a square BHI agar plate and placed in an upright position to allow the drops to run down the plate to facilitate counting of the colonies. To select donor cells, BHI agar contained 10 $\mu$g/ml tetracycline and 25 $\mu$g/ml fusidic acid, and to select for transconjugant cells, BHI agar contained 10 $\mu$g/ml tetracycline and 20 $\mu$g/ml erythromycin. The plates were incubated for ~24 h at 37°C before colonies were counted and enumerated for colony-forming units (CFU). The frequency of DNA transfer is presented as the number of transconjugants per donor. Experiments were done in triplicates and are reported with their SD.

For the serial passaging, conjugation assays were performed in the exponential phase as described above. Three colonies of the transconjugant plates from passage 1 were picked to start new

overnight cultures, which were then used as donor cells for the following passage. In passage 2, donor cells were therefore OG1ES: pCF10, and OG1RF without a plasmid served as recipient cells. Three transconjugant colonies from passage 2 served as donor cells for passage 3 with OG1ES as recipient cells, and transconjugant cells from passage 3 were donors for passage 4 with OG1RF as recipient cells. Donor and transconjugant cells were selected as previously described for passages 1 and 3. For passages 2 and 4, BHI agar containing 10 µg/ml tetracycline and 20 µg/ml erythromycin was used to select for donor cells and BHI agar containing 10 µg/ml tetracycline and 25 µg/ml fusidic acid was used to select for transconjugants.

All in vivo data are from three biological replicates and are plotted with their SD using GraphPad Prism (version 10.2) (GraphPad Software). Statistical significance was analyzed with one-way ANOVA.

## Data Availability

Atomic coordinates and structure factors have been deposited with the Protein Data Bank with the accession codes 8S4S and 8S4T for the apo and DNA-bound structures, respectively.

## Supplementary Information

## Acknowledgements

The authors would like to thank Dr. Krishna Chaitanya Bhattiprolu, Dr. Lena Lassinantti, and Dr. Saba Shahzad for input on PcfG production and purification. We also thank Dr. Josy ter Beek for rewarding discussions regarding the project. We thank the Chemical Biology Consortium Sweden (CBCS), and we acknowledge Protein Production Sweden (PPS) for providing facilities and experimental support. PPS is funded by the Swedish Research Council as a national research infrastructure. We acknowledge MAX IV Laboratory for time on Beamline BioMax under Proposal 20180236. Research conducted at MAX IV, a Swedish national user facility, is supported by the Swedish Research Council under contract 2018-07152, the Swedish Governmental Agency for Innovation Systems under contract 2018-04969, and Formas under contract 2019-02496. We also acknowledge the synchrotron ESRF (France) for time at beamlines ID23 and ID30. This work was supported by grants from the Swedish Research Council (2016-03599 and 2023-02423 to RP-A Berntsson, 2019-01874 to PH Wanrooij), Knut and Alice Wallenberg Foundation (to each RP-A Berntsson and PH Wanrooij), and Kempestiftelserna (SMK-1762 and SMK-1869 to RP-A Berntsson).

### Author Contributions

A Breidenstein: formal analysis, validation, investigation, visualization, and writing—original draft, review, and editing.
A Lamy: conceptualization, formal analysis, and investigation.
CPJ Bader: validation, investigation, and visualization.
W-S Sun: validation and investigation.
PH Wanrooij: supervision, funding acquisition, methodology, and writing—review and editing.
RP-A Berntsson: conceptualization, supervision, funding acquisition, investigation, visualization, project administration, and writing—original draft, review, and editing.

### Conflict of Interest Statement

The authors declare that they have no conflict of interest.

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
