## [Reviewer comments · Life Science Alliance]

Life Science Alliance

PrgE from plasmid pCF10 is an OB-fold protein that is strikingly different from bacterial SSBs

Annika Breidenstein, Anais Lamy, Cyrielle Bader, Wei-Sheng Sun, Paulina Wanrooij, and Ronnie Berntsson
DOI: <https://doi.org/10.26508/lsa.202402693>

Corresponding author(s): Ronnie Berntsson, Umeå University

Review Timeline:

Submission Date:	2024-03-05
Editorial Decision:	2024-04-08
Revision Received:	2024-05-03
Editorial Decision:	2024-05-06
Revision Received:	2024-05-08
Accepted:	2024-05-08

Transaction Report:

April 8, 2024

Re: Life Science Alliance manuscript #LSA-2024-02693

Dr. Ronnie P-A Berntsson
Umeå University
Department of Medical Biochemistry and Biophysics
Umeå 901 87
Sweden

Dear Dr. Berntsson,

Thank you for submitting your manuscript entitled "PrgE from plasmid pCF10 is an OB-fold protein that is strikingly different from bacterial SSBs" to Life Science Alliance. The manuscript was assessed by expert reviewers, whose comments are appended to this letter. We invite you to submit a revised manuscript addressing the Reviewer comments.

Thank you for this interesting contribution to Life Science Alliance. We are looking forward to receiving your revised manuscript.

Sincerely,

B. MANUSCRIPT ORGANIZATION AND FORMATTING:

Reviewer #1 (Comments to the Authors (Required)):

This study presents a comprehensive analysis of PrgE, an OB-fold protein encoded by the conjugative plasmid pCF10, encompassing structural, biochemical and functional aspects.

The authors conducted a thorough investigation of PrgE, by analyzing its oligomerization state, its structure in the absence or presence of DNA, and its role in conjugation.

The data presented here suggests that this OB-fold protein may exhibit a distinct role or mode of action compared to classical SSBs but still retains some common characteristics with SSB from viral particles.

The paper is nicely written and hypothesis are supported by the performed experiments.

No major comments.

Some minor comments and suggestions:

- Figure 3B: remove nM from the Y-axis label.
- Figure 3B: Please verify if the protein concentrations used for SEC analysis are in μM rather than nM?
- Figure 3C: it would be interesting to show the mass distribution along the elution peak, thus we could estimate the mass of delayed complexes.
- Regarding PrgE role in conjugation, it will be interesting to test filter mating which was showed previously that it may give different results compared to liquid mating.
- Superimpositions of PrgE-DNA structure with SSBs-DNA complexes in Fig. 5 do not clearly show the differences. Perhaps using different colors for DNA will help to improve the comparison.
- It is interesting to know of this family of OB-fold proteins is widely distributed among conjugative plasmids

Reviewer #2 (Comments to the Authors (Required)):

This manuscript reports on the crystal structure of PrgE, which is encoded by the *Enterococcus faecalis* plasmid pCF10 from within the transfer (*tra*) operon. The authors show that PrgE carries an OB-fold characteristic of SSBs but has different properties from known bacterial or eukaryotic SSBs. A major difference is that PrgE assembles on single-stranded DNA as beads on a string, very much unlike the paradigmatic *E. coli* SSB which assembles as a tetramer around which the ssDNA wraps. Additionally, PrgE binds both ssDNA and dsDNA, apparently without sequence specificity. PrgE's role as an SSB is not known, therefore the authors favor designating it as an OB-fold protein with properties distinctly different from other OB-fold SSBs. Some similarities are noted with viral OB-fold proteins, which leads to the suggestion that PrgE was acquired from a viral or phage source. Overall, the experiments are well-crafted, including the use of complementary approaches to decipher structure and function, and the manuscript is written in a way that is engaging to the audience. The discovery of an OB-fold protein with novel oligomerization and DNA binding properties should attract a wide readership. The authors should address the following:

1. L. 72. The phrase "with corresponding molecular weights" should be deleted or more fully described.
2. L. 84. The reference 15 isn't appropriate here, it's a review.
3. L. 126. A reference or URL should be provided here.
4. L. 248. Pray should be prey.
5. L. 301. Although it is mentioned later, it would also be good here to mention the other major difference relating to the presence of distinctive α -helices that are not shared by the chromosomal SSB. These are more obvious structural variations than is evident for the OB-fold, and it is also relevant that the AlphaFold model and x-ray structure agree on their presence/structure.
6. L. 344 and 356. In view of the oligomeric states and the structure showing that PrgE binds ssDNA as beads on a string, it would be interesting to determine if the deletion of the N-terminal tail impacts both oligomerization and ssDNA binding properties. Not a critical experiment for this paper, but the structure certainly suggests this tail is an important feature for the observed ssDNA binding architecture.
7. In the context of the above and Fig. S4, A complement to the fluorescence anisotropy experiments would be gel shift assays with increasing concentrations of PrgE to test for cooperative binding. Given the nature of the N-terminal tail interactions with adjacent monomers, it is surprising - at least to this reviewer - that Prg doesn't bind cooperatively.

8. L. 402. The dsDNA binding affinity is interesting and could suggest a different function than the proposed SSB function. Given that the protein binds ssDNA and dsDNA equally well, why favor the SSB model over, for example, a role in transcriptional regulation or replication or cell division, etc. Other proteins, such as HU or IHF or HN-S bind dsDNA as chains, and have such functions. Why not propose the same for PrgE, even though it does not share the same structural features of these NAPs? Are the functions known for the viral OB-fold proteins shown to bind ssDNA and dsDNA equally well? If so, such functions could serve as a guide here, at least for development of models to be tested in the future.

9. L. 402. Although a structure of PrgE bound to dsDNA would be nice, based on the PrgE-ssDNA structure, it seems possible that PrgE could induce bending of dsDNA - can this be tested?

10. L. 433 and 437. Does a deletion mutant of *E. faecalis* SSB exist? If so, it would be interesting to see whether *E. faecalis* can complement for the chromosomal SSB - or even the *E. coli* *ssb* deletion.

11. Have the authors searched the *E. faecalis* genome for PrgE-like proteins, or proteins with OB-folds that might function redundantly to PrgE?

12. Fig. 7. It might be useful to show that PrgF and PcfG pull each other down, providing at least some sort of a positive control that these proteins were purified in a form in which they could bind partners. Also, again, prey should be prey.

Reviewer #3 (Comments to the Authors (Required)):

Review report "PrgE: an OB-fold protein from plasmid pCF10 with striking differences to prototypical 1 bacterial SSBs" by Breidenstein et al.

This interesting paper describes a rather detailed study on the presumed single-stranded DNA binding protein (SSB) protein encoded by the *Enterococcus faecalis* conjugative plasmid pCF10. All organisms encode an SSB protein, and also most if not all conjugative plasmids encode (at least) one SSB. So far, however, the only SSBs studied were those encoded by conjugative plasmids of Gram negative bacteria. Based on this, this paper is therefore interesting. In addition, the results are extensive and include the surprising result that PrgE can bind dsDNA with similar K_d as to ssDNA. The conclusions reached in the paper are sustained by the presented data. I enjoyed reading the paper; the results are clearly described and presented in an organized manner. As it stands, the paper merits publication in life science alliance, but its impact can be improved considerably by providing three additional results. One of these is answering the question whether PrgE can complement the host-encoded SSB, second is studying the role of the N-terminal tail (not obtaining the crystal structure of the mutant complexed to ssDNA, but to know its effect on ss and dsDNA binding activity). And finally, the crystal structure of PrgE bound to dsDNA. Answering the last question can be a study on its own, but perhaps answering (one of) the first two questions are feasible.

A list of other major and minor comments is given below.

Major comments:

Introduction. Conjugative DNA is transferred through the T4SS in the recipient as ssDNA. I believe this is rather important to mention this in the introduction.

Discussion, The authors may comment whether PrgE remains in the donor or it is transferred to the recipient cells (does contain features suggestive for transfer by the the T4SS?).

L319. Why was PrgE crystallized in 1:3 molar ration (with polyA 60-mer ssDNA)?

L356: The N-terminal tail of one PrgE binds to its neighbour, indicating it has an important role for binding to ssDNA (and perhaps dsDNA). It would be very interesting to know how a N-terminal deletion variant would affect binding to ss and/or dsDNA.

L404. Binding to dsDNA was tested using a 30 or 60 nt molecules. For ssDNA it is specified that the DNA tested was polyA. What is the nature of dsDNA? Have different dsDNA molecules been tested, like AT-rich compared to GC-rich. Make a comment here or in discussion.

L414 and further, Essentiality of PrgE for conjugation was tested. The paper would gain importance by knowing if PrgE can complement the *E. faecalis* SSB.

As mentioned above, the authors obtained the intriguing result that PrgE can bind dsDNA. Obtaining the structure of PrgE bound to dsDNA can be a study on itself, but they may speculate in the discussion how PrgE would bind to dsDNA, perhaps including binding of other OB-fold ssDNA binding protein that also binds to dsDNA 8as stated in the paper), if this is known.

Minor comments:

L70: at end add for example " . . . in the second DNA binding mode"

Lines 73-74: not fluent, improve sentence (. . . have more in common . . .)

L81, remove "also"

L296 and Fig S1: show in Fig 1 or Fig S1 the primary sequence of PrgE, Indicate the length of PrgE and SSB of *E. coli* and *S.*

feacalis. Indicate in Fig S1 N- and C-terminus. .

L306. Tone down "indicates" to "suggests"

L314. Fig S2, explain colours (are they as in Fig 1?) and/or indicate N and C-terminus

L319, consider removing "for PrgE"

L371, typo (dot)

L397, in legend Fig 5D-F mention that PrgE-ssDNA is shown in brown.

L523. Italic *E. coli*

Response to reviewers

We would like to thank the reviewers for their thoughtful comments and suggestions, which have helped us to further strengthen the manuscript and improve the clarity. Based on the suggestions, we have now included additional data on a N-terminal deletion variant of PrgE, which required some reorganization of the manuscript (we moved the results on oligomerization after the data on the DNA-bound crystal structure) and renumbering of figures. Please find our point-by-point response to all reviewer comments below (reviewer comments in black, our response in green text).

Reviewer #1 (Comments to the Authors (Required)):

This study presents a comprehensive analysis of PrgE, an OB-fold protein encoded by the conjugative plasmid pCF10, encompassing structural, biochemical and functional aspects. The authors conducted a thorough investigation of PrgE, by analyzing its oligomerization state, its structure in the absence or presence of DNA, and its role in conjugation. The data presented here suggests that this OB-fold protein may exhibit a distinct role or mode of action compared to classical SSBs but still retains some common characteristics with SSB from viral particles.

The paper is nicely written and hypothesis are supported by the performed experiments. No major comments.

Some minor comments and suggestions:

- Figure 3B: remove nM from the Y-axis label.

Thank you for catching this mistake. We have now corrected nM to nm in the Y-axis labels.

- Figure 3B: Please verify if the protein concentrations used for SEC analysis are in μM rather than nM?

We initially indicated the amount of PrgE in nmol in the figure to account for elution over different volumes, however in the figure legend we referred to protein concentration, which was unclear. We now show the concentration of the injected protein in the figure (in μM) to avoid this confusion.

- Figure 3C: it would be interesting to show the mass distribution along the elution peak, thus we could estimate the mass of delayed complexes.

The mass distribution along the entire elution peak approaches that of a monomer (with a theoretical molecular weight of 17 kDa), see figure below. However, while the calculated mass is very accurate at the center of the peak, it's much less so in the rest of the run. The SEC-MALS analysis requires alignment of the different signals (UV,

light scattering, refractive index) at the peak, which means that the error gets larger when you move away from the aligned peak. If we increase and show the calculated mass along the entire peak the technical error of the measurement increases to more than 10%, which makes it difficult to draw any strong conclusions from the data.

[Figure removed by editorial staff per authors' request].

- Regarding PrgE role in conjugation, it will be interesting to test filter mating which was showed previously that it may give different results compared to liquid mating.

The main advantage of filter mating is that it leads to a higher conjugation efficiency than liquid mating, which can help determine differences in situations with low conjugation transfer rates (transconjugants/donor). However, both wt pCF10 and pCF10: $\Delta prgE$ have very high (and similar) efficiencies, and then filter mating would not make it easier to discern any differences. It has previously been shown that there is virtually no difference between filter mating and liquid mating for wt pCF10 (see eg. Laverde Gomez *et al.* PrgK, a Multidomain Peptidoglycan Hydrolase, Is Essential for Conjugative Transfer of the Pheromone-Responsive Plasmid pCF10. *J. Appl. Microbiol.* **196**, 527–539 (2014).

- Superimpositions of PrgE-DNA structure with SSBs-DNA complexes in Fig. 5 do not clearly show the differences. Perhaps using different colors for DNA will help to improve the comparison.

Thank you for the suggestion. It has been slightly challenging to make these figures as clear as possible. We have now used darker tones of the same colors for DNA to improve the contrast between DNA and protein, and feel that this improves clarity.

- It is interesting to know of this family of OB-fold proteins is widely distributed among conjugative plasmids

We agree that this would be very interesting. However, it is difficult to differentiate between genomic and plasmid encoded protein sequences that are available in the NCBI database. For a related study, we are developing a database of T4SS containing plasmids. This dataset indicates that PrgE is not widely distributed, but since this dataset and the coupled methodology has not yet been validated, we cannot say this for certain. We hope to revisit this question in the future, but since it requires quite a bit of work, we feel that it falls outside the scope of this article. However, we have expanded the introductory results section to include the additional information that PrgE homologs can only be found in Lactobacillales, as deduced by primary sequence blasts (lines 112-114).

Reviewer #2 (Comments to the Authors (Required)):

This manuscript reports on the crystal structure of PrgE, which is encoded by the *Enterococcus faecalis* plasmid pCF10 from within the transfer (*tra*) operon. The authors show that PrgE carries an OB-fold characteristic of SSBs but has different properties from known bacterial or eukaryotic SSBs. A major difference is that PrgE assembles on single-stranded DNA as beads on a string, very much unlike the paradigmatic *E. coli* SSB which assembles as a tetramer around which the ssDNA wraps. Additionally, PrgE binds both ssDNA and dsDNA, apparently without sequence specificity. PrgE's role as an SSB is not known, therefore the authors favor designating it as an OB-fold protein with properties distinctly different from other OB-fold SSBs. Some similarities are noted with viral OB-fold proteins, which leads to the suggestion that PrgE was acquired from a viral or phage source. Overall, the experiments are well-crafted, including the use of complementary approaches to decipher structure and function, and the manuscript is written in a way that is engaging to the audience. The discovery of an OB-fold protein with novel oligomerization and DNA binding properties should attract a wide readership. The authors should address the following:

1. L. 72. The phrase "with corresponding molecular weights" should be deleted or more fully described.

We deleted this phrase.

2. L. 84. The reference 15 isn't appropriate here, it's a review.

We apologize for this, and have now removed the reference here.

3. L. 126. A reference or URL should be provided here.

We refer to the company making the LEX system here, namely Epiphyte3, which is common practice for instruments.

4. L. 248. Pray should be prey.

Thank you for catching this honestly embarrassing typo. We have now changed it throughout the manuscript.

5. L. 301. Although it is mentioned later, it would also be good here to mention the other major difference relating to the presence of distinctive α -helices that are not shared by the chromosomal SSB. These are more obvious structural variations than is evident for the OB-fold, and it is also relevant that the AlphaFold model and x-ray structure agree on their presence/structure.

Thank you for the suggestion, we now refer to the additional α -helices already here in the description of the AlphaFold model (lines 122-123). We write about the agreement between the AlphaFold model and the crystal structure in the next paragraph (lines 143-144).

6. L. 344 and 356. In view of the oligomeric states and the structure showing that PrgE binds ssDNA as beads on a string, it would be interesting to determine if the deletion of the N-terminal tail impacts both oligomerization and ssDNA binding properties. Not a critical experiment for this paper, but the structure certainly suggests this tail is an important feature for the observed ssDNA binding architecture.

Thank you for the suggestion. We have now created an N-terminal deletion variant of PrgE (Δ N-PrgE) where we removed the first 12 residues. This variant could readily be purified, but did indeed behave differently to wild-type PrgE, both in oligomerization and in DNA binding. Δ N-PrgE is mostly a monomer in solution, but can dimerize in low salt (as determined by SEC-MALS). It binds to DNA with lower affinity than wild-type PrgE. We have now added this data to the manuscript, and rewritten the corresponding paragraphs (lines 223-254), as well as new Fig. 6 and Fig. 7 and Fig. S4C and updated the discussion on lines 304-314.

7. In the context of the above and Fig. S4, A complement to the fluorescence anisotropy experiments would be gel shift assays with increasing concentrations of PrgE to test for cooperative binding. Given the nature of the N-terminal tail interactions with adjacent monomers, it is surprising - at least to this reviewer - that Prg doesn't bind cooperatively.

We have extensively performed EMSAs, but were unable to get properly reproducible results. Typically, we observed smeared bands with higher protein concentrations, but they were not of high enough quality to be included in the manuscript.

However, we agree with your observation on the cooperativity and we therefore reanalyzed our fluorescence anisotropy data of PrgE and Δ N-PrgE with the Hill equation that can take potential cooperativity into account. For most of the data, the Hill equation does not fit as well as the quadratic equation we used previously, suggesting that PrgE does not bind cooperatively. However, for the 60-mer ssDNA a positive Hill constant (ca 1.5) gives the best fit in both 50 and 100 mM NaCl, suggesting cooperative binding at least for longer DNA substrates. This cooperativity disappeared for Δ N-PrgE, which fits well with what can be expected from the crystal structure and oligomerization. We have now updated the results to include this data on cooperativity (lines 244-254), and added a new panel to Fig. S4, as well as updated the discussion (lines 309-314).

8. L. 402. The dsDNA binding affinity is interesting and could suggest a different function than the proposed SSB function. Given that the protein binds ssDNA and dsDNA equally well, why favor the SSB model over, for example, a role in transcriptional regulation or replication or cell division, etc. Other proteins, such as HU or IHF or HN-S bind dsDNA as chains, and have such functions. Why not propose the same for PrgE, even though it does not share the same structural features of these NAPs? Are the functions known for the viral OB-fold proteins shown to bind ssDNA and dsDNA equally well? If so, such functions could serve as a guide here, at least for development of models to be tested in the future.

We primarily discuss PrgE in the context of SSBs due to previous literature and its OB-fold, which is primarily associated with SSB function. It is very much possible that PrgE has another function, but since we do not have any data to point us in a specific direction, we prefer not to go into those speculations in depth. To make this clearer, we have now modified the discussion to emphasize that PrgE might be involved in any process involving DNA (lines 373-375).

9. L. 402. Although a structure of PrgE bound to dsDNA would be nice, based on the PrgE-ssDNA structure, it seems possible that PrgE could induce bending of dsDNA - can this be tested?

We agree that it would be very nice to have a structure of PrgE bound to dsDNA. However, despite trying we have unfortunately been unable to determine that structure. Traditional DNA bending experiments using EMSAs requires a specific binding sequence. We have previously studied DNA compaction using nanochannels, via collaboration, which could in theory answer the question. However, those experiments were not straightforward to set up and we feel that this falls outside the scope of the current work.

10. L. 433 and 437. Does a deletion mutant of *E. faecalis* SSB exist? If so, it would be interesting to see whether *E. faecalis* can complement for the chromosomal SSB - or even the *E. coli* ssb deletion.

Unfortunately, we do not possess a deletion mutant of the *E. faecalis* SSB. We agree that it would be interesting to test, although we think that it is unlikely that PrgE would be able to complement for genomically encoded *E. faecalis* SSB due to the differences we describe in Fig. S1.

11. Have the authors searched the *E. faecalis* genome for PrgE-like proteins, or proteins with OB-folds that might function redundantly to PrgE?

We have done a variety of searches using BLAST and Foldseek to find clues about a potential function of PrgE, but they were not very insightful. Blasting PrgE against the *E. faecalis* strain OG1RF that we used in our study resulted only in one hit with 100% identity, which is PrgE itself. Using Foldseek to search the AlphaFold database (AFDB50) only found hits with low sequence identity and the top hits were uncharacterized proteins.

Based on this comment and other reviewer suggestions, we have expanded the introductory results paragraph where we describe this in the manuscript (lines 112-129).

12. Fig. 7. It might be useful to show that PrgF and PcfG pull each other down, providing at least some sort of a positive control that these proteins were purified in a form in which they could bind partners. Also, again, pray should be prey.

We performed the experiments in a similar manner as was described in a previous study from us (Rehman et al, 2019), there we have shown that PcfF indeed pulls down PcfG. This is referenced in the manuscript. Further indication that the proteins are functional is that PcfG is functional and nicks DNA as has also been previously described (Chen *et al.* Specificity determinants of conjugative DNA processing in the *Enterococcus faecalis* plasmid pCF10 and the *Lactococcus lactis* plasmid pRS01. *Mol. Microbiol.* **63**, 1549–1564 (2007)).

We have also repeated these nicking experiments our selves and also in our hands PcfG is functional (see nicking assay below).

[Figure removed by editorial staff per authors' request].

Reviewer #3 (Comments to the Authors (Required)):

Review report "PrgE: an OB-fold protein from plasmid pCF10 with striking differences to prototypical bacterial SSBs" by Breidenstein et al.

This interesting paper describes a rather detailed study on the presumed single-stranded DNA binding protein (SSB) protein encoded by the *Enterococcus faecalis* conjugative plasmid pCF10. All organisms encode an SSB protein, and also most if not all conjugative plasmids encode (at least) one SSB. So far, however, the only SSBs studied were those encoded by conjugative plasmids of Gram negative bacteria. Based on this, this paper is therefore interesting. In addition, the results are extensive and include the surprising result that PrgE can bind dsDNA with similar K_d as to ssDNA. The conclusions reached in the paper are sustained by the presented data. I enjoyed reading the paper; the results are clearly described and presented in an organized manner. As it stands, the paper merits publication in life science alliance, but its impact can be improved considerably by providing three additional results. One of these is answering the question whether PrgE can complement the host-encoded SSB, second is studying the role of the N-terminal tail (not obtaining the crystal structure of the mutant complexed to ssDNA, but to know its effect on ss and dsDNA binding activity). And finally, the crystal structure of PrgE bound to dsDNA. Answering the last question can be a study on its own, but perhaps answering (one of) the first two questions are feasible.

A list of other major and minor comments is given below.

Major comments:

Introduction. Conjugative DNA is transferred through the T4SS in the recipient as ssDNA. I believe this is rather important to mention this in the introduction.

Thank you for this suggestion, we agree. We added a clearer description of the DNA transfer in the introduction (lines 55-58).

Discussion, The authors may comment whether PrgE remains in the donor or it is transferred to the recipient cells (does contain features suggestive for transfer by the the T4SS?).

This is indeed an interesting question. We currently do not know if the pCF10 T4SS can transfer proteins other than the relaxase to recipient cells, as has been shown for SSB on the F plasmid. We have now included text along these lines in the discussion (lines 365-368).

L319. Why was PrgE crystallized in 1:3 molar ration (with polyA 60-mer ssDNA)?

At the time when we performed the crystallization trials, our working hypothesis was that PrgE predominantly is trimeric and we thought that it might wrap ssDNA around said trimer in a way similar to *E. coli* SSB. We therefore chose a 1:3 molar ratio and a 60-mer as the approximate length that *E. coli* SSB binds. We chose polyA instead of a random sequence to facilitate model building.

L356: The N-terminal tail of one PrgE binds to its neighbour, indicating it has an important role for binding to ssDNA (and perhaps dsDNA). It would be very interesting to know how a N-terminal deletion variant would affect binding to ss and/or dsDNA.

Thank you for this suggestion, which was very similar to points 6 and 7 from reviewer #2. Please see our response to those questions.

L404. Binding to dsDNA was tested using a 30 or 60 nt molecules. For ssDNA it is specified that the DNA tested was polyA. What is the nature of dsDNA? Have different dsDNA molecules been tested, like AT-rich compared to GC-rich. Make a comment here or in discussion.

We used poly A DNA only for crystallization, not for binding assays, as we considered random DNA a more relevant substrate. We have now clarified this in the results and added a reference to Table S2 where the sequences can be found (line 237).

We designed the random DNA sequence with <https://www.generi-biotech.com/oligoapp/> and roughly equal amounts of AT-GC. The sequence is the same for ss and dsDNA and the 30-mer corresponds to one half of the sequence of the 60-mer.

L414 and further, Essentiality of PrgE for conjugation was tested. The paper would gain importance by knowing if PrgE can complement the *E. faecalis* SSB.

We agree that this would indeed be very interesting to test. Unfortunately, making knock-outs in *E. faecalis* is sometimes very difficult, for reasons that are not completely clear, and at this stage we do not have an SSB knock-out strain available to us.

As mentioned above, the authors obtained the intriguing result that PrgE can bind dsDNA. Obtaining the structure of PrgE bound to dsDNA can be a study on itself, but they may speculate in the discussion how PrgE would bind to dsDNA, perhaps including binding of other OB-fold ssDNA binding protein that also binds to dsDNA as stated in the paper), if this is known.

As mentioned in our response to point 9 from reviewer #2, we have tried to crystallize PrgE with dsDNA, but have unfortunately not been successful.

Analyzing the ssDNA-bound structure, it seems clear that there are differences in how PrgE binds ss- and dsDNA, as there is not enough space to accommodate dsDNA in the crystal structure. Other available structures of OB-folds binding dsDNA (eg PDB code 4L5S) shows a different mode of binding than PrgE to ssDNA, with the DNA in a different structural conformation. We have identified interactions to the ssDNA phosphate backbone, and these interactions might also be involved in binding to dsDNA, but the ds DNA must have a different structure than what we observe in the ssDNA-bound structure. This makes it difficult to speculate exactly how PrgE would bind dsDNA. We have added this kind of reasoning to the discussion, lines 322-325.

Minor comments:

L70: at end add for example " . . . in the second DNA binding mode"

Fixed

Lines 73-74: not fluent, improve sentence (. . . have more in common . . .)

We have now rephrased this part.

L81, remove "also"

Fixed.

L296 and Fig S1: show in Fig 1 or Fig S1 the primary sequence of PrgE, Indicate the length of PrgE and SSB of *E. coli* and *S. faecalis*. Indicate in Fig S1 N- and C-terminus. .

Thank you for this suggestion, we have now included N- and C-terminus labels in the

figure. We also added a panel to Fig S1, in which we show a sequence alignment of the primary sequences of PrgE, *E. faecalis* SSB and *E. coli* SSB. We think the alignment is a useful complement to show the similarities and differences between the proteins.

L306. Tone down "indicates" to "suggests"

Done.

L314. Fig S2, explain colours (are they as in Fig 1?) and/or indicate N and C-terminus

We added a description of the colours in the figure legend.

L319, consider removing "for PrgE"

Done

L371, typo (dot)

Done

L397, in legend Fig 5D-F mention that PrgE-ssDNA is shown in brown.

Done

L523. Italic *E. coli*

Done.

May 6, 2024

RE: Life Science Alliance Manuscript #LSA-2024-02693R

Dr. Ronnie P-A Berntsson
Umeå University
Department of Medical Biochemistry and Biophysics
Umeå 901 87
Sweden

Dear Dr. Berntsson,

Thank you for submitting your revised manuscript entitled "PrgE from plasmid pCF10 is an OB-fold protein that is strikingly different from bacterial SSBs". We would be happy to publish your paper in Life Science Alliance pending final revisions necessary to meet our formatting guidelines.

- please be sure that the authorship listing and order is correct
- please add callouts for Figure 7 panels A-D, and for Figure S4D

A. FINAL FILES:

B. MANUSCRIPT ORGANIZATION AND FORMATTING:

****It is Life Science Alliance policy that if requested, original data images must be made available to the editors. Failure to provide original images upon request will result in unavoidable delays in publication. Please ensure that you have access to all original**

data images prior to final submission.**

The license to publish form must be signed before your manuscript can be sent to production. A link to the electronic license to publish form will be available to the corresponding author only. Please take a moment to check your funder requirements.

Sincerely,

May 8, 2024

RE: Life Science Alliance Manuscript #LSA-2024-02693RR

Dr. Ronnie P-A Berntsson
Umeå University
Department of Medical Biochemistry and Biophysics
Umeå 901 87
Sweden

Dear Dr. Berntsson,

Thank you for submitting your Research Article entitled "PrgE from plasmid pCF10 is an OB-fold protein that is strikingly different from bacterial SSBs". It is a pleasure to let you know that your manuscript is now accepted for publication in Life Science Alliance. Congratulations on this interesting work.

DISTRIBUTION OF MATERIALS:

Again, congratulations on a very nice paper. I hope you found the review process to be constructive and are pleased with how the manuscript was handled editorially. We look forward to future exciting submissions from your lab.

Sincerely,
